# Transcriptome Analysis Reveals Candidate Genes Involved in Gibberellin-Induced Fruit Development in *Rosa roxburghii*

**DOI:** 10.3390/plants12193425

**Published:** 2023-09-28

**Authors:** Xiaolong Huang, Xiaoai Wu, Guilian Sun, Yu Jiang, Huiqing Yan

**Affiliations:** 1School of Life Sciences, Guizhou Normal University, Guiyang 550001, China; huangxiaolong@gznu.edu.cn (X.H.); wuxiaoai1234@163.com (X.W.); sungl2022@163.com (G.S.); abc20090820@163.com (Y.J.); 2Key Laboratory of Plant Physiology and Development Regulation, Guizhou Normal University, Guiyang 550001, China; 3Laboratory of State Forestry Administration on Biodiversity Conservation in Mountainous Karst Area of Southwestern China, Guizhou Normal University, Guiyang 550001, China

**Keywords:** *Rosa roxburghii*, gibberellin, transcriptome, fruit development

## Abstract

Gibberellins (GAs) play indispensable roles in the fruit development of horticultural plants. Unfortunately, the molecular basis behind GAs regulating fruit development in *R. roxburghii* remains obscure. Here, GA_3_ spraying to *R. roxburghii* ‘Guinong 5’ at full-bloom promoted fruit size and weight, prickle development, seed abortion, ascorbic acid accumulation, and reduction in total soluble sugar. RNA-Seq analysis was conducted to generate 45.75 Gb clean reads from GA_3_- and non-treated fruits at 120 days after pollination. We obtained 4275 unigenes belonging to differently expressed genes (DEGs). Gene ontology and the Kyoto Encyclopedia of Genes and Genomes displayed that carbon metabolism and oxidative phosphorylation were highly enriched. The increased critical genes of DEGs related to pentose phosphate, glycolysis/gluconeogenesis, and citrate cycle pathways might be essential for soluble sugar degradation. Analysis of DEGs implicated in ascorbate revealed the myoinositol pathway required to accumulate ascorbic acid. Finally, DEGs involved in endogenous phytohormones and transcription factors, including R2R3 MYB, bHLH, and WRKY, were determined. These findings indicated that GA_3_-trigged morphological alterations might be related to the primary metabolites, hormone signaling, and transcription factors, providing potential candidate genes that could be guided to enhance the fruit development of *R. roxburghii* in practical approaches.

## 1. Introduction

Fruit development of horticultural plants is a refined reproductive and consecutive process divided into three stages: fruit set, the cell division or expansion stage, and fruit maturation [1]. The transition from flowering to fruit set upon fertilization in angiosperm refers to the initiation of fruit development. It begins with a period of cell division, and its high cell division activities often rapidly take place within a few weeks inside the ovary after anthesis and mitotic cells can still be found [2]. The second phase lasts a long time, mainly determining the final fruit size and weight. The last period is fruit thorough maturation and the fruit size grows very slightly at this stage. However, multiple storage products and secondary metabolites accumulate, thus forming different intrinsic and extrinsic qualities of fruits that are accompanied and could be manipulated by several phytohormones [3].

Gibberellins (GAs) are phytohormones and play critical roles in all the physiological periods of horticultural fruit development [4]. Abundant GAs are perceived as signals by the ovary or other fruit tissues after fertilization, resulting in fruit cell expansion [2]. For example, enhancing expression levels of genes in the gibberellin synthesis positively regulates the fruit set of tomato plants (*Solanum lycopersicum*) [4,5]. Similarly, endogenous GA accumulation is consistent with high cell division levels and tomato fruit expansion [6]. Moreover, GA application in the woodland strawberry (*Fragaria vesca*) promotes longitudinal fruit elongation, increasing the fruit size and weight [7]. Likewise, an early study proved that exogenous spraying of GAs increased the fruit weight of apples (*Malus × domestica*) and the wax layer thickness to decrease the fruit water-loss rate. It also lowers the proportion of asymmetrical fruit [8]. Transient overexpression of *PbGA20ox2* promoted the fruit development of pear trees (*Pyrus bretschneideri*) and delayed the drop of nonpollinated fruit [4]. Beyond that, GA treatment induced parthenocarpy (fertilization-independent fruit) in horticultural fruits, such as cucumbers and grapes [9,10]. All conclusions prove that GAs play essential roles in fruit development.

Recent studies focus on GA signaling and how it determines the biological functions in horticultural fruits [11]. When bioactive GAs are increased, a GA-binding pocket in GID1 (GIBBERELLININ SENSITIV EDWARF1) captures bioactive GAs. Simultaneously, DELLAs bind with the GID1, forming a GID1-GA-DELLA complex, and are subsequently degraded by the 26S proteasome, releasing repressed multiple transcription factors (TFs) to regulate transcripts of corresponding downstream genes, for example, MYB, bHLH, NAC, and TCP could be activated by GAs [12,13]. These candidate TFs exert their potential functions on fruit development [13]. In addition, exogenous GAs trigger diverse and complex crosstalk with other endogenous phytohormones. For example, GA-signaling repressor SlDELLA binds with auxin signaling components SlIAA9 and SlARF7 by yeast two-hybrid assays and coimmunoprecipitation, thereby repressing fruit initiation and inducing the parthenocarpy of tomatoes [2]. Arabidopsis DELLA also interacts with auxin-responsive factors (AtARF6, AtARF7, and AtARF8) to regulate hypocotyl elongation [14]. Gibberellin stimulates polar auxin transport and has a common transcriptome with auxin in the development of Populus wood [15]. Gibberellic acid (GA_3_) application also results in the promotion of histidine kinase (*AHK*) in the cytokine signaling pathway, which positively regulates the initiation of axillary meristem in garlic (*Allium sativum*) [16]. Therefore, the signaling pathways or other phytohormones induced by GAs potentially influence the fruit development of horticultural plants.

*Rosa roxburghii* Tratt is also known as a chestnut rose belonging to the Rosaceae family. It is a deciduous, thick shrub that produces prickles on its stems and fruits [17]. Its fruits are widely consumed in China for their pleasant flavor and high nutrients. They are even dubbed the king of vitamin C, since they provide an excellent L-ascorbic acid source [18]. Some studies prove that phytohormones regulate the seedling growth and fruit quality of *R. roxburghii*. For example, a low concentration (1 mM) of methyl jasmonate (MeJA) elicited the root growth of *R. roxburghii* and promoted triterpenoid synthesis [19]. Exogenous GA treatment was performed on the fruits of *R. roxburghii* in vitro [20]. Unfortunately, information regarding the effect of GAs on fruit development in vivo and the molecular mechanisms behind it are still unavailable. Advances in high-throughput sequencing technologies enable the comprehensive analysis of deep mechanisms, providing an opportunity to understand the complex genetic networks and molecular basis underlying the effects of GAs [21,22,23]. However, extensive transcriptome alterations in response to exogenous GA_3_ application in *R. roxburghii* fruits remain scarce.

In this current study, we used Illumina sequencing technology to conduct RNA-Seq transcriptome of *R. roxburghii* fruits with and without GA_3_ application. Subsequent comparison of the global expression profiles of GA_3_-treated and untreated *R. roxburghii* fruits allowed the identification of numerous GA_3_-responsive genes. This study aims to explore these mechanisms through a robust transcriptome analysis of gibberellin-treated fruits, providing several potential candidate genes to promote the fruit quality of *R. roxburghii*.

## 2. Results

### 2.1. Exogenous Gibberellin Applications Result in Physiological Changes in R. roxburghii Fruits

To explore the roles of GA_3_ on the fruit development of *R. roxburghii*, flowers of seven-year-old seedlings in full blossom were treated with GA_3_ (200 mg/L) daily for 7 days. We observed that the fruit treated with GA_3_ was more significant than the mock at both 60 days after pollination (DAP) or 120 DAP (Figure 1A). Accordingly, GA_3_ induced a 13.61% increase in fresh fruit weight and a 21.79% increase in fruit shape index when fruits matured at 120 DAP, suggesting that GA_3_ substantially positively regulated the size of *R. roxburghii* fruit cells (Figure 1B). Interestingly, the prickle length of the fruit increased by about a third, reaching 33.59%. In contrast, the seed number decreased by 31.23% after GA_3_ treatment. Considering that total soluble sugars (TSS) (sucrose, glucose, and fructose) and L-ascorbic acid were critical qualities of *R. roxburghii* fruits, we measured their biochemical traits as shown in Figure 1B. GA_3_ application resulted in a significant decrease in TSS but increased L-ascorbic acid concentration. Hence, GA_3_ significantly influenced the fruit development of *R. roxburghii*, including regulating fruit weight and size, prickle, seed failure, and fruit quality.

### 2.2. Transcriptome Sequencing and De Novo Assemble of R. roxburghii

To better explore the molecular mechanism underlying the physiological changes by GA_3_, we used RNA-seq to profile the transcriptomes of GA_3_-treated and non-treated fruits at 120 DAP. High-throughput sequencing generated a total of 45.75 Gb clean reads for six samples, and the percentage of Q30 bases ranged from 94.59% to 95.11% (Appendix A). The clean reads were de novo assembled into transcripts, and 32,895 unigenes were obtained with an average length of 1294 bp (N50 = 1933). A length of 23,395 (71.11%) unigenes was over 500 bp, while those of 15,437 (46.92%) unigenes were over 1 Kb, suggesting that the assembly quality of the *R. roxburghii* transcriptome was satisfactory (Figure 2A). A total of 26,294 unigenes were annotated according to at least one database. Among them, 93.85% of (24,677) unigenes were annotated based on the National Center for Biotechnology Information (NCBI) non-redundant (Nr) database. In terms of the Nr annotations, 40.24% of the sequences had strong homology (E-value < 1 × 10^−150^), 11.96% and 20.71% of the annotated sequences showed homology (1 × 10^−150^ < E-value < 1 × 10^−100^), and (1 × 10^−100^ < E-value < 1 × 10^−50^) to available plant sequences, respectively (Figure 2B). The similarity distribution was comparable, with 41.56% of the sequences having similarities higher than 90%, while 48.66% of the sequences of *R. roxburghii* had similarities of 50–90% (Figure 2C). Regarding homolog species, Nr annotated 54.09% of unigenes which had top matches to sequences from woodland strawberries (*Fragaria vesca* subsp), followed by *Prunus persica* (2.03%), *Prunus avium* (1.48%), *Prunus mume* (1.15%), *Pyrus×bretschneideri* (0.99%), *Malus domestica* (0.88%), and others (39.17%), suggesting that *Fragaria vesca* was the outstanding species that shared significantly high homologous to *R. roxburghii* (Figure 2D).

### 2.3. Transcriptome-Scale Analysis of GA_3_-Responsive DEGs in R. roxburghii Fruits

Differentially expressed genes (DEGs) were analyzed using the Fragments Per Kilobase of the exon model per million mapped fragments (FPKM) to assess the degree of overlap between GA_3_- and non-treated fruits. A total of 4275 DEGs were detected, with 2782 up-regulated and 1513 down-regulated genes (Figure 3A). Among all DEGs, 89.47% (3825) unigenes were annotated using at least one database (Appendix A).

Many DEGs annotated within the biological process category of Gene ontology (GO) were analyzed (Figure 3B). For instance, most genes were identified to play essential roles in metabolic, cellular, and single-organism processes, biological regulation, response to stimulus, and signaling. In addition, they were also found to be involved in other multiple biological processes, including developmental, multicellular organismal, and reproductive processes (Appendix A). We also evaluated DEGs based on their cellular components and their molecular functions. Regarding the cellular component category, most DEGs fell within the cell part, membrane, organelle, macromolecular complex, and membrane-enclosed lumen. As for molecular function classification, the highest abundance of DEGs was related to catalytic activity and binding. At the same time, other fascinating groups, including transporter, structural molecule, and transcription factor activities, were clustered. Moreover, most DEGs also participated in molecular function regulator, electron carrier, antioxidant, and signal transducer activities. Clusters of Orthologous Groups of proteins (COG) assignment was performed to classify the functions of DEGs (Figure 3C). GA_3_ treatment significantly regulated the expression of numerous genes encoding carbohydrate transport and metabolism, indicating their potential roles in the TSS accumulation in *R. roxburghii* fruits after GA_3_ application. Other COG functions triggered by GA_3_ were translation, ribosomal and biogenesis, energy production and conversion, and secondary metabolites biosynthesis.

The Kyoto Encyclopedia of Genes and Genomes (KEGG) enrichment analysis can identify the major biochemical and signal transduction pathways in which the DEGs were involved. The significant entries of DEGs were implicated in the carbon metabolism, oxidative phosphorylation, citrate cycle, glycolysis/gluconeogenesis, phenylpropanoid biosynthesis, and biosynthesis of amino acids (Figure 4A). DEGs associated with ascorbate, aldarate, tyrosine, tryptophan, starch, and sucrose metabolism were also enriched (Figure 4B).

### 2.4. DEGs Related to Primary Metabolism after GA_3_ Application in R. roxburghii Fruits

Considering that the TSS accumulation of *R. roxburghii* fruits decreased significantly after exogenous GA_3_ treatment, we determined the DEGs concerning primary metabolism. Their pathways are roughly outlined in Figure 5. The KEGG enrichments suggested that 52, 44, and 15 DEGs were associated with glycolysis/gluconeogenesis, citrate cycle (TCA), and the pentose phosphate pathway, respectively. Regarding the glycolysis/gluconeogenesis pathway, the expression profiles of DEGs displayed two clusters. Among these, 32.69% of DEGs (17) were significantly enhanced in fruits after gibberellin application. For example, two unigene-encoded phosphoenolpyruvate carboxykinase were significantly raised. Moreover, we found that unigenes encoding pyruvate kinase, pyruvate dehydrogenase E1 component, fructose–bisphosphate aldolase, glyceraldehyde-3-phosphate dehydrogenase, dihydrolipoyl dehydrogenase, and glyceraldehyde 3-phosphate dehydrogenase were also substantially increased, suggesting that these DEGs might play critical roles in decomposing soluble sugar. (Appendix A). Moreover, about half of DEGs related to the pentose phosphate pathway showed a similar tendency after GA_3_ treatment. Among these, the expression levels of unigenes annotated as 6-phosphogluconate dehydrogenase and transaldolase were promoted. Likewise, 43.19% of DEGs involved in the TCA pathway were significantly enhanced, especially ATP citrate (pro-S)-lyase, malate dehydrogenase, and phosphoenolpyruvate carboxykinase. Hence, some candidate crucial genes in the primary metabolic pathway by exogenous GA_3_ in *R. roxburghii* fruits were identified, indicating that they might play crucial roles in degrading the total soluble sugar of *R. roxburghii* fruits.

### 2.5. Ascorbate and Hormone-Related Signaling Pathways upon Exogenous GA_3_ Applications

The proposed L-ascorbic acid synthetic and recycling pathways are shown in Figure 6A. We identified and analyzed candidate DEGs implicated in ascorbate metabolism. The expression levels of DEGs encoding biosynthesized ascorbate were remarkably increased, mainly including UDP-glucose 6-dehydrogenase and inositol oxygenase (Figure 6B). L-ascorbic acid can be oxidized to ascorbate catalyzed by ascorbate oxidase and reversely by monodehydroascorbate reductase (NADH) activities. Despite the decrease in unigenes encoding these two enzymes, the reduction in unigenes annotated as L-ascorbate oxidase is more significant than the expression of monodehydroascorbate reductase, resulting in the accumulation of L-ascorbic acid in *R. roxburghii* fruits (Appendix A).

To investigate the roles of other endogenous hormones in *R. roxburghii* fruits, genes associated with hormone signaling pathways were identified upon GA_3_ treatment (Figure 6C). Among them, DEGs were mainly involved in five principal plant hormones: auxin, cytokinins (CK), salicylic acid (SA), abscisic acid (ABA), and ethylene (Appendix A). DEGs encoding AUXIN/Indole-3-acetic acid (AUX/IAA), Small auxin-up RNA (SAUR), and Gretchen Hagen 3 (GH3) were found, supporting the crosstalk of the GA_3_ signaling pathway with the auxin signaling pathway in *R. roxburghii*. Given that AUX/IAA proteins repressed auxin signaling, the decrease in DEGs annotated as AUX/IAA suggested that GA_3_ spraying enhanced the auxin signaling pathway. Likewise, histidine-containing phosphotransferase protein 1-like (*RrAHP*) was stimulated to regulate cell division. By contrast, DEGs related to the signaling pathways of salicylic acid, abscisic acid, and ethylene were reduced, indicating that exogenous GA_3_ may be conducive to alleviating the inhibitory effect of ABA and ethylene signal transduction that is commonly known to inhibit the growth of specific tissues and cells. These results indicated that exogenous GA_3_ applications could affect other hormone-related pathways in *R. roxburghii*.

### 2.6. DEGs Related to Transcription Factors in R. roxburghii Fruits

The gene expression of transcription factors influenced by gibberellin in *R. roxburghii* fruits was investigated. A total of 577 candidate TFs were annotated in the transcriptome of *R. roxburghii* fruits. We obtained 24 TFs with significant differences (Appendix A). TF members comprised the MYB family (MYB82 and R2R3 MYB), WRKY family (WRKY41 and WRKY75), bZIP family, MADS, NAC, and others. The expression levels of *TGA_3_*, *C2H2*, *R2R3 MYB*, and *WRKY* were significantly increased (Figure 7). These TFs might play significant roles in regulating the development of *R. roxburghii* fruit, providing information for studying the functions of TFs in the promoting effect of GA_3_ on the fruit quality of *R. roxburghii*.

### 2.7. Validation of DEGs by qRT-PCR

To verify the reliability of the transcriptome results, we randomly selected nine genes in the related pathways of *R. roxburghii* fruits for qRT-PCR validation, including unigenes involved in L-ascorbate acid, sugar, hormone signaling, and key transcriptional factors. Primers were listed (Appendix A), and their expression levels were calculated using the 2^−ΔΔCt^ method. We further compared the expression data of the DEGs obtained by RNA-seq and qRT-PCR, and the results displayed a similar tendency (Figure 8A). The correlation between RNA-Seq (FPKM) and qPCR (2^−ΔΔCt^) results for the nine DEGs was also calculated using log_2_fold variation measurements to produce a scatter plot. The qRT-PCR results of nine DEGs were significantly similar to the RNA-seq results (R^2^ = 0.7598), indicating that our RNA-seq data were accurate and reproducible (Figure 8B).

## 3. Discussion

### 3.1. Exogenous Gibberellin Spraying Regulates Fruit Qualities of R. roxburghii

The GA application in *R. roxburghii* induced the apparent physiological divergences of the fruits, including promoting fruit size and weight, ascorbic acid accumulation, and prickle development. The increase in fruit size was consistent with other horticultural fruits, such as grapes, apples, and tomatoes [7]. Interestingly, the development of fruit prickle was also promoted by GA_3_, the same as the functions of increased wax layer thickness by GA_3_ treatment in apples, which could reduce fruit water loss rate [8]. Meanwhile, the increase in ascorbic acid and prickle length accompanied by the reduction in TSS concentration was noticed, which might be aroused by the flux allocation between primary and secondary metabolism [24]. Moreover, we observed that the seed number was significantly decreased, possibly due to fertilization failure, sharing the same phenomenon of parthenocarpy (fertilized-independent fruit) that could be induced by gibberellin [4]. Similarly, overexpression of GA biosynthesis enzymes could significantly decrease the seed number of tomatoes [25]. Likewise, CRISPR/Cas9-generated knockout GA biosynthesis enzyme mutants increased the seed numbers of soybeans (*Glycine max*), suggesting that GA_3_ application negatively regulated seed set [26].

### 3.2. GA_3_ Facilitated the Decomposition of Total Soluble Sugars by increasing Crucial Genes in Primary Metabolism

In this current study, six DGE libraries were constructed using RNA-Seq and used to screen DEGs after GA_3_ treatment. A total of 26,294 unique sequences were obtained, of which 24,677 unigenes were annotated. The results of the present study would be helpful to clone coding domain sequences and analyze gene families in *R. roxburghii*. Multiple genes were altered during fruit setting after GA_3_ treatment through RNA-Seq analysis. The roles of many annotated genes showed the same plant responses with GAs in other species. For example, *c241716.graph_c1* was annotated as an expansion, which was also increased in *Malus domestica* after the spray of gibberellin [27]. Moreover, GA_3_ significantly decreases the transcript of genes encoding UDP-glycosyltransferase in *R. roxburghii*, as in *F. vesca* [28]. We analyzed GO and KEGG pathways and determined that 3825 DEGs participated in several pathways, yielding novel comprehensive insights into the GA_3_ response in *R. roxburghii* fruits.

A few transcriptional events of GAs drive drastic shifts at the primary metabolites and developmental events [29]. We found that starch and sucrose metabolism pathways were significantly enriched, similar to the metabolic pathway that GAs induced during the fruit setting of triploid Loquat (*Eriobotrya japonica*) [21]. The RNA-seq data displayed that multiple DEGs involved in the glycolytic pathway, TCA, and pentose phosphate pathway were up-regulated, indicating that GAs accelerated the decomposing of storage sugars [30]. Spraying GA_3_ reduced TSS contents in *R. roxburghii* fruits, mainly because of the enhanced DEGs in the primary metabolisms. A similar tendency was found that the sugar contents were also reduced in germinated seeds of *Fraxinus hupehensis* treated with GAs [30]. Most enzymes were encoded by gene families that displayed various expression levels in *R. roxburghii*. However, many enzymes might be increased after GA application by considering expressions of all genes in the same family [31]. For example, glyceraldehyde-3-phosphate dehydrogenase (GAPDH), a key enzyme in the glycolytic pathway that catalyzes the conversion of D-glyceraldehyde 3-phosphate to 1,3-diphosphoglycerate, was annotated by four genes. The increase in one unigene (*c266210.graph_c0*) is more extensive than the sum of the other four genes (*c252200.graph_c0*, *c252200.graph_c1*, *c265626.graph_c0*, and *c265626.graph_c1*) (Appendix A). Therefore, GA_3_ facilitated the oxidative decomposition of soluble sugars by increasing crucial genes in primary metabolism.

### 3.3. The Myoinositol Pathway Was Required for the Accumulation of Ascorbic Acid in R. roxburghii

The abundance of ascorbic acid was influenced by the biosynthesis and recycling pathways [32]. GA_3_ spraying could significantly enhance the concentration of ascorbic acid. Moreover, the RNA-seq determined that DEGs implicated in the biosynthesis of L-ascorbate were all impressively enhanced, especially UDP-glucose 6-dehydrogenase and inositol oxygenase, possibly suggesting that L-galactose and myoinositol pathways were essential for the accumulation of ascorbic acid. Meanwhile, exogenous GA_3_ substantially decreased the expression of L-ascorbate oxidase, increasing the reduced content and redox state. The results would help the understanding of the molecular basis regulating ascorbic acid in *R. roxburghii* fruits.

### 3.4. The Altered Hormonal Signaling and TF Induced by GA_3_ Were Responsible for the Fruit Development of R. roxburghii

Transcriptome analyses illustrated that other hormone signaling pathways showed enhanced sensitivity to the application of GA_3_. Five auxin-related DEGs were identified. *c246759.graph_c0* encoding (AUX/IAA), which could bind to auxin response factors, was significantly down-regulated (Appendix A) [33]. An early study also proved that silencing of tomato *SlIAA9* led to the parthenocarpic fruit of Solanaceae [34]. Moreover, two genes encoding GH3 were decreased. Since GH3 was reported to catalyze the synthesis of indole-3-acetic acid (IAA)-amino acid conjugates, the reduction in GH3 indicated a high level of free IAA concentration [35]. During the early periods of fruit expansion, the transcript of GH3 was at a lower level, according to the high auxin concentration at the same development stage in apples [36]. Hence, GAs may play a role in increasing auxin signaling to regulate the fruit development of *R. roxburghii*. Similarly, Illumina HiSeq high-throughput sequencing revealed that auxin-responsive genes, playing the same roles of *c254917.graph_c0* in *R. roxburghii*, also increased in *F. vesca* when treated with 50 mg/L GA_3_ [37]_._ In addition, histidine-containing phosphotransfer protein 1-like involved in the cytokinin signaling pathway was promoted after GA_3_ treatment, indicating that GAs possibly induced fruit physiological changes via the cytokinin signal pathway [38]. ABA inhibits fruit growth in the early stages of strawberries, accompanied by low ABA levels in early development and a sharp increase during ripening [7]. Our study showed the decreased DEGs implicated in the ABA signaling pathway. Morever, some DEGs involved in other phytohormones were also determined. These results confirmed that auxin, ethylene, abscisic acid, and salicylic acid, which GAs can directly or indirectly trigger, might regulate fruit development, similar to the complex hormonal regulation in tomatoes [39].

Based on DGE analysis, the expression levels of the genes encoding several transcription factors were significantly changed during GA_3_ treatment in *R. roxburghii*, including genes encoding NAC, MADS, bHLH, MYBs, and others. TFs play crucial roles in fruit qualities. For example, the development of *R. roxburghii* fruit prickles is determined by the complex of MYB-bHLH-WD40. Interestingly, the prickle length was significantly enhanced by GA application in our study. The same treatment also induced trichome formation by MYB (GLABROUS 1) in Arabidopsis [40]. RrGL1 (MYB), RrGL3/EGL3 (bHLH), and RrTTG1 (WD40) are responsible for prickle development [41,42]. The two former TFs are required for prickle elongation. In contrast, WD40 is necessary for inducing prickle initiation. Interestingly, The RNA-seq analyzed the transcripts of genes encoding *R2R3 MYB* and *bHLH041* that were significantly up-regulated, showing that the conserved domains with *RrGL1* and *RrGL3/RrEGL3* might play similar effects on prickle development. However, *RrTTG1* displayed no significant change in *R. roxburghii*, which explained why prickle only became longer but not numerous [17].

## 4. Materials and Methods

### 4.1. The Plant Materials

Seedlings of the *Rosa roxburghii* Tratt cultivar ‘Guinong 5’ were planted at the garden of Guizhou Normal University in Guiyang. The trees were 7 years old, in the adult phase. Flowers of *R. roxburghii* in full blossom were sprayed with 200 mg/L of GA_3_ (Sigma-Aldrich, cat# Sigma G7645, St. Louis, MI, USA) daily for 7 days as treatment and distilled H_2_O as a control on 15 May 2022 (Appendix A). We collected at least twenty fruits at 60 DAP and 120 DAP for morphology analysis. Some fresh fruits treated with GA_3_ and control at 120 DAP were harvested from three separate seedings for measurement. The others for RNA extraction were snap-frozen immediately in nitrogen and stored in a −80 °C refrigerator. Transcriptome sequencing was performed with three biological replications.

### 4.2. Plant Biomass Yield and Biochemical Analysis of R. roxburghii

Physiological data of at least twenty fruits from three separate seedlings were recorded on the following traits: average fruit weight (g), fruit shape index (calculated as longitudinal diameter (cm)/horizontal diameter (cm)), seed number, and prickle length per each fruit. We weighed 100 g of fruits per treatment and homogenized it with 5000 rpm. Then, 2 g of the sample was collected, powdered with liquid nitrogen with 20 mL ethanol (80%, *v*/*v*), and sonicated for 15 min at 80 °C. After evaporation, the residue was dissolved with 2 mL of distilled water for HPLC analysis. The percentages (g/g) of total soluble sugar (TSS) and L-ascorbic acid concentration (g/g) were determined using high-performance liquid chromatography (HPLC) as described by Huang et al. [31].

### 4.3. RNA Extraction, Library Construction, and Sequencing of R. roxburghii

Total RNA was extracted from *R. roxburghii* fruits using the TRIzol reagent (Invitrogen, Carlsbad, CA, USA). RNA quality and quantity were determined using a NanoDrop spectrophotometer and Agilent 2100 Bioanalyzer for further transcriptome sequencing. RNA (1 μg) per sample was used for the preparations. Following the manufacturer’s recommendations, sequencing libraries were generated using NEBNext^®^Ultra™ RNA Library Prep Kit for Illumina^®^ (NEB, Ipswich, MA, USA). The library preparations were sequenced on an Illumina Hiseq 2000 platform. Clean data were obtained by removing reads containing adapter, ploy-N, and low-quality reads from raw data. All the RNA-seq raw data have been uploaded and are available in Genbank under accession number PRJNA1003688.

### 4.4. De Novo Assembles and Unigenes Annotation of R. roxburghii

Transcriptome assembly was accomplished based on the left. fq and right. fq using Trinity with min_kmer_cov set to 2 by default and all other parameters set to default [43]. Gene functions were annotated and based on BLASTx with cutoff E-value ≤ 1 × 10^−5^ and HMMER ≤ 1 × 10^−10^ [44] using Nr, GO [45], COG [46], and KEGG [47] databases.

### 4.5. Differential Expression Analysis of Unigenes Expressed in R. roxburghii Fruits

The expression level was estimated with RNA-Seq by Expectation Maximization (RSEM). FPKM values indicate the abundance of the corresponding unigene [48]. DESeq R package (1.10.1) was adopted to compare the GA_3_ treatment with the reference group [49]. The resulting *p* values were adjusted using Benjamini and Hochberg’s approach for controlling the false discovery rate (FDR). An adjusted *p*-value < 0.001 and |log_2_(fold change)| ≥ 2 were set as the threshold for determining the DEGs. GO and KEGG pathway analyses were conducted to interpret the DEGs.

### 4.6. Real-Time Quantitative PCR

Total RNAs from the *R. roxburghii* fruits were isolated, and RT-PCR Kit^®^ (TaKaRa, Shiga, Japan) was performed using 2 μg of RNA and oligo dT-adaptor primer. Real-time quantitative PCR was performed in a LightCycler480 instrument (Roche, Basel, Switzerland). Each reaction contained 10 μL of SYBR green PCR master mix (TaKaRa, Shiga, Japan), 1.0 μL cDNA, 200 nM primers, and ddH_2_O up to the final volume of 20 μL. Amplification was performed at 95 °C for 5 min, followed by 40 cycles at 95 °C for 20 s, 58 °C for 30 s, and 72 °C for 1 min. The expression levels relative to the RrActin were estimated using the 2^−ΔΔCt^ method. All samples were performed for three biological replicates.

### 4.7. Statistical Analysis

The results were statistically evaluated by one-way analysis of variance (ANOVA) followed by Tukey’s test using Statistical Program for Social Sciences (SPSS) program version 20.0 (SPSS Inc., Chicago, IL, USA). Statistical differences with *p* values under 0.05 are considered significant, using an asterisk that indicates a significant difference. “*”: *p* < 0.05; “**”: *p* < 0.01.

## 5. Conclusions

Our study reveals a complex network of genes regulated by GA_3_ treatment in *R. roxburghii* fruits. The enriched GO terms and KEGG pathways highlight the multifaceted roles of GAs in promoting cell growth, accelerating fruit ripening, and modulating stress responses. These findings provide valuable insights into the molecular mechanisms of GA action in *R. roxburghii* fruits and provide a foundation for future studies to improve fruit quality and yield. Although significant DEGs have been identified regarding the roles of GAs in *R. roxburghii*, many problems remain to be studied. Efficient genetic transformation and plant regeneration systems are still lacking for *R. roxburghii*, limiting the functional validation of gene studies on relevant studies. It is worth constructing transgenetic systems in *R. roxburghii* in the future, which will undoubtedly promote further understanding of the molecular mechanisms underlying the effects of GAs on fruit development in *R. roxburghii*.

## Figures and Tables

**Figure 1 plants-12-03425-f001:**
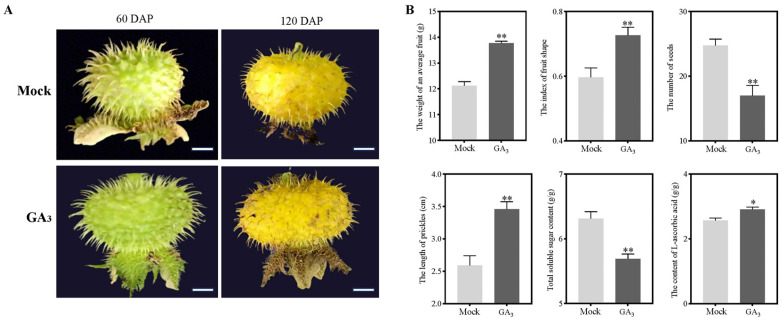
Exogenous gibberellin applications influenced the fruit development of *R. roxburghii*. (**A**) The morphology of fruits at 60 DAP (days after pollination) and 120 DAP after gibberellin (GA_3_) spraying. Bar = 1 cm. (**B**) Effects of GA_3_ on fruit weight, fruit shape index, seed number, prickle length, concentration of total soluble sugar, and L-ascorbic acid contents. Values were Mean ± S. D. from at least twenty fruits. The asterisk indicates a significant difference. “*”: *p* < 0.05; “**”: *p* < 0.01.

**Figure 2 plants-12-03425-f002:**
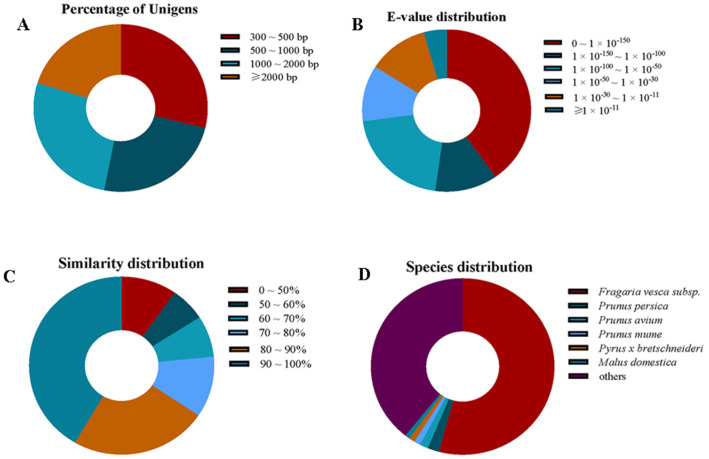
Characterization of the unigenes of *R. roxburghii* fruit transcriptome. (**A**) Statistics of the length distribution of assembled unigenes. (**B**) E-value distribution of the homology search of unigenes against the non-redundant (Nr) database. (**C**) Similarity distribution of the unigenes for each unique sequence. (**D**) Nr annotated species distribution similar to *R. roxburghii* fruit transcriptome.

**Figure 3 plants-12-03425-f003:**
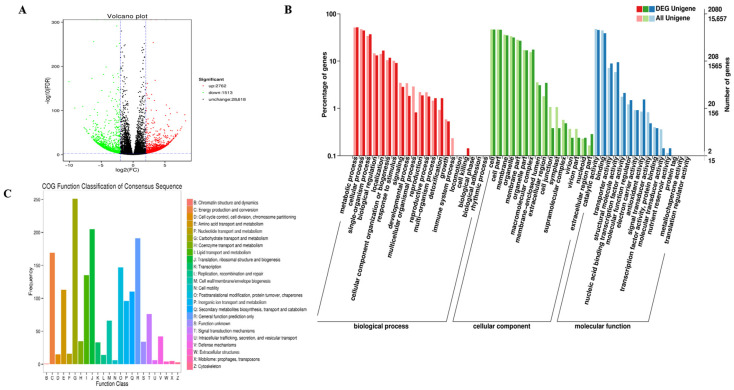
Global analysis of differentially expressed genes in response to gibberellins. (**A**) Volcano plot of DEGs expressed genes with the cutoff (|log2(fold change)| ≥ 2 and an adjusted *p*-value < 0.001). (**B**) Most significant GO functions (**C**) COG function annotations of DEGs expressed in *R. roxburghii* fruits.

**Figure 4 plants-12-03425-f004:**
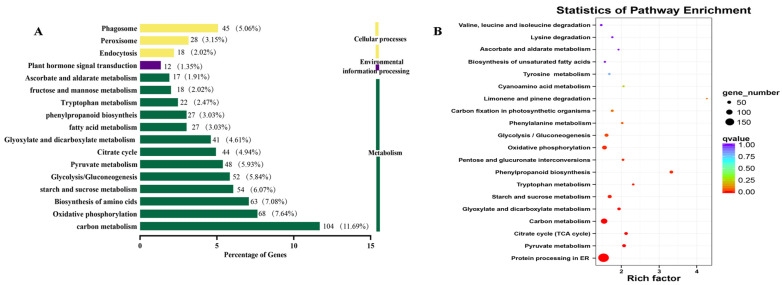
KEGG pathway enrichment of DEGs. (**A**) Statistical analysis of annotated genes in KEEG pathways, (**B**) scatterplot of KEEG pathway for DEGs.

**Figure 5 plants-12-03425-f005:**
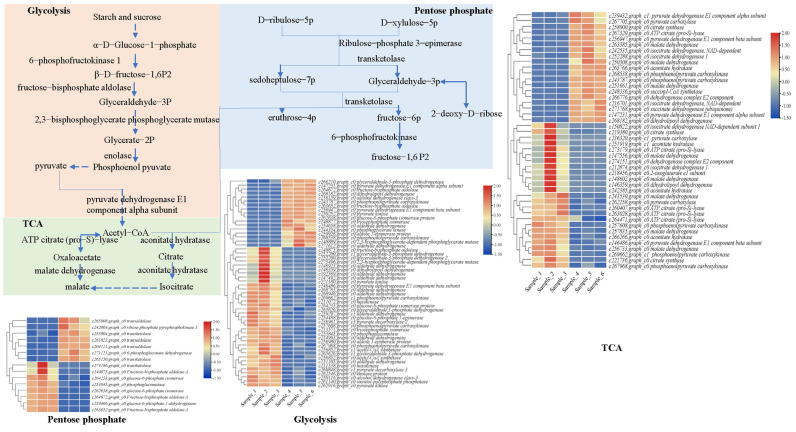
Expression profiles of DEGs related to primary metabolism of *R. roxburghii* fruit. The pathways of glycolysis/gluconeogenesis, citrate cycle (TCA), and pentose phosphate pathways were outlined and expression levels of candidate DEGs were listed using a heatmap. Bars represent the scale of the FPKM of each gene, as indicated by the red and blue rectangles.

**Figure 6 plants-12-03425-f006:**
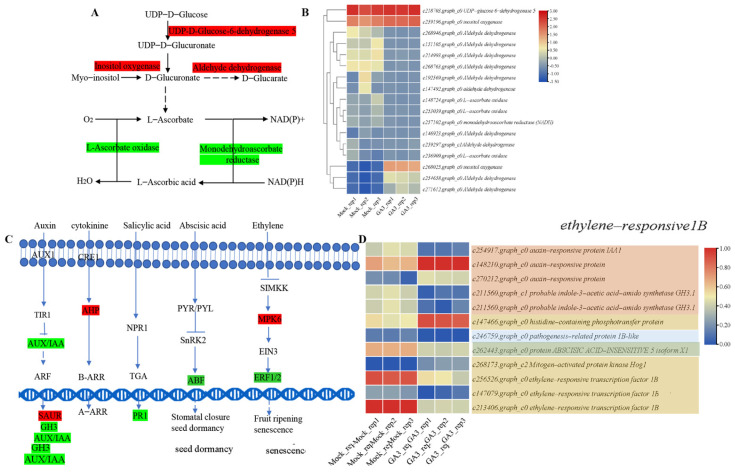
Expression profiles of DEGs related to ascorbate metabolism and hormone signaling of *R. roxburghii* fruit. (**A**) The DEGs involved in the ascorbate pathway are listed. (**B**) Heatmap of DEGs-influenced ascorbate. (**C**) The simple sketch of hormone signaling pathways. (**D**) Heatmap of DEGs implicated in endogenous hormone signaling pathways. The enzymes in the red shadow represent up-regulated, whereas those in the green shadow indicate down-regulated.

**Figure 7 plants-12-03425-f007:**
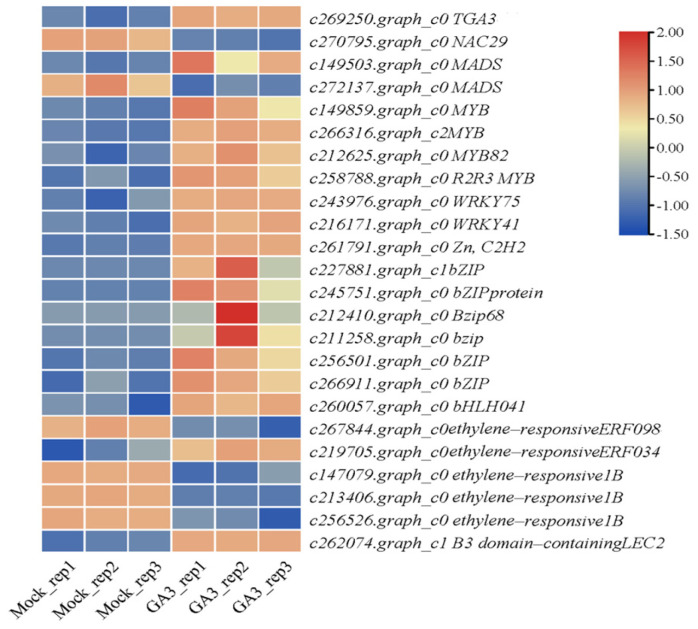
GA_3_ promoted the expression levels of most transcription factors.

**Figure 8 plants-12-03425-f008:**
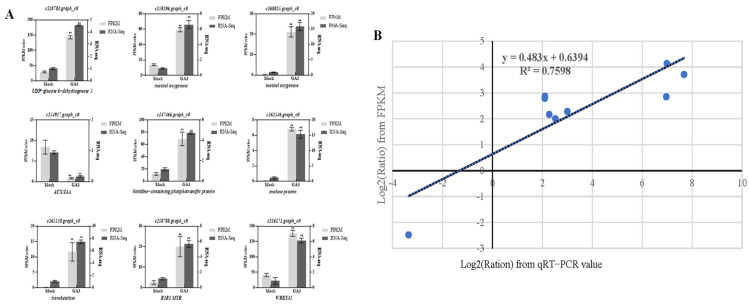
The qRT-PCR verification of DEGs. (**A**) The boxplot reveals the comparison of expression levels by RNA-seq and qRT-PCR. (**B**) Correlation analysis of qPCR (2^−ΔΔCt^) and RNA-Seq results (RPKM). Log_2_fold change of RNA-seq (*y*-axis) and qRT-PCR (*x*-axis) were listed. The asterisk indicates a significant difference. “**”: *p* < 0.01.

## Data Availability

The RNA-Seq of raw data is publicly available in the NCBI under number: PRJNA1003688 with the link: https://www.ncbi.nlm.nih.gov/sra/PRJNA1003688 accessed on 11 August 2023.

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
