# Peer review of "Transcriptome Analysis Reveals Candidate Genes Involved in Gibberellin-Induced Fruit Development in Rosa roxburghii"

_plants, 2023, doi:10.3390/plants12193425_

Round 1

Reviewer 1 Report

The research paper highlights the role of Gibberellins (GAs) in fruit development in horticultural plants, particularly Rosa roxburghii. The study investigates the effects of GA3 application on various aspects of fruit development, including size, weight, prickle development, seed abortion, ascorbic acid accumulation, and soluble sugar content. Transcriptome sequencing was conducted to unravel the molecular basis underlying these responses. Key findings include the identification of a big number of candidate genes associated with primary metabolism, hormone signaling, and transcription factors, offering potential insights into enhancing fruit development in R. roxburghii through GA3 application. The results are very preliminary to provide valuable insights into the molecular mechanisms of GA action in R. roxburghii fruits and provide a foundation for future studies to improve fruit quality and yield.

Specific Comments:

Absrtact

It is suggested to write the abstract to make it shorter and more effective in highlighting the most interesting aspects of the research. In particular it is suggested to check the meaning of the sentence in line 12

Methods

Details about the number of replicates for Physiological data recorded should be included.

Author Response

Dear Reviewer:

We would like to express our special thanks for your carefully editing our manuscript entitled “Transcriptome analysis reveals candidate genes involved in gibberellin-induced fruit development in Rosa roxburghii” (ID: 2596896). The comments are precious and helpful. We have studied the comments carefully and made corresponding modifications that meet the requirements for publication in Plants. Besides, thank you for your summary. We would like to express our special thanks for your time and for carefully editing our manuscript. 

Here, the responses to the comments are listed as follows:

Abstract It is suggested to write the abstract to make it shorter and more effective in highlighting the most interesting aspects of the research. In particular it is suggested to check the meaning of the sentence in line 12

Response: We sincerely thank you for the kind comment. We have made some reductions to the abstract to present the substantial and interesting results of this study. Besides, to avoid ambiguous and redundant meanings, we have deleted the previous sentence in Line 12.

Methods

Details about the number of replicates for Physiological data recorded should be included.

Response: Really thanks for your critical review. We have added the number of replicates in the revised manuscript. At least twenty fruits from three separate 7-year-old seedlings were collected for physiological data.

Finally, many thanks for your careful review. We appreciate your time in reviewing our manuscript during this time. Our deepest gratitude goes to you for your careful work and thoughtful suggestions that have helped improve this paper substantially. We wish good health to you, and your family. 

Best regards,

Sincerely,

Huiqing

Reviewer 2 Report

This study is written regarding how gibberellin spraying to R. roxburghii affects fruit development and gene expression and it might provide new approach to how we can utilize phytohormone to maximize fruit yield increase. However, I feel it is required to revise to improve the quality of a manuscript.

My comments are as follows:

L83 Italic

L88 Can you please a bit mention how low concentration of MeJA elicited root growth and triterpenoids synthesis induce seedling growth and fruit quality? 

L88 Methyl jasmonate?

L106 (200 mg/L)

L106 7 days

L125 I am unsure if this word is right here. Please confirm again.

L134 Can you please provide what NCBI is for readers?

L141 Are there any known roles regarding these genes on growth promotion and is there consensus of plant responses with GAs among different species (including R. roxburghii)? If so, please document in the discussion part.

L279 Discussion part should be written by each results by making sub-headings. 

L289 Can you please more case studies of how GAs affect seed numbers in chestnut roses and/or other flowering plants?

L374 Italic

L375 7

L377 7 days

L378 May 15 2023?

L378 How many samples per treatments?

L383 I recommend changing title such as 'Plant biomass yield and biochemical analysis of R roxburghii'

L384 Traits:

L385 I recommend providing a formula.

L386 100g per each sample or treaments? and homogenized with centrifuge with which rpm?

L398 Unify format (company, region, country)

L408 (RNA-Seq by Expectation-Maximization)Avoid abbreviation at the first time. 

L426 SPSS version 20.01, is it right?

I believe there are no substantial issues in the quality of English writing. 

Round 2

Reviewer 2 Report

I think the authors mostly reflected my comments and tried to improve the quality of the manuscript, but there are some comments regarding the required revision process. Please find the attachment. Thank you

English writing seems fine but it might need some proofreading for the final step.
